# Activated Carbon for Sepsis Prevention and Intervention: A Modern Way of Utilizing Old Therapies

Elisaveta Snezhkova [1,*], Heinz Redl [2,3], Johannes Grillari [2,3,4] and Marcin Osuchowski [2,*]

1   Department of Means & Methods of Adsorptive Therapy, R. E. Kavetsky Institute of Experimental Pathology, Oncology and Radiobiology, National Academy of Sciences of Ukraine, 03022 Kyiv, Ukraine
2   Ludwig Boltzmann Institute for Traumatology, The Research Center in Cooperation with AUVA, LBG, Donaueschingenstrasse 13, 1200 Vienna, Austria; heinz.redl@trauma.lbg.ac.at (H.R.); johannes.grillari@trauma.lbg.ac.at (J.G.)
3   Austrian Cluster for Tissue Regeneration, 1200 Vienna, Austria
4   Institute of Molecular Biotechnology, University of Natural Resources and Life Sciences, Muthgasse 18, 1190 Wien, Austria
*   Correspondence: lisasne@hotmail.com (E.S.); marcin.osuchowski@trauma.lbg.ac.at (M.O.)

**Abstract:** (1) Background: Uncontrolled inflammation often contributes to life-threatening sepsis sequela such as multi-organ dysfunction syndrome (MODS), and is accompanied by abnormal levels of pathological and damage-associated molecular patterns (PAMPs & DAMPs) in biological fluids. Activated carbon or charcoal (AC) of new generation with ameliorated biocompatibility has spurred renewed interest in the regulation of these toxins' levels in inflammation states. (2) Methods: We searched PubMed, Google Scholar, ScienceDirect, Researchgate, and other sources for the relevant literature from 1550 B.C. till 2022 A.C. (3) Results: Laboratory and clinical investigations demonstrate that activated carbon or charcoal (AC) mitigates inflammation in different pathological states when applied orally, per rectum, or in a hemoperfusion system. AC protects the microbiome and bone marrow, acts as an anti-inflammatory and anti-oxidant remedy, and recovers the plasmatic albumin structure. The mechanism of AC action is related to a non-selective (broad-range) or/and selective adsorption of PAMPs & DAMPs from biological fluids. A high-adsorptive capacity towards noxious substances and application of AC as early as possible seems paramount in inflammation treatment for preventing sepsis and/or multi-organ failure. (4) Conclusion: AC could be considered an adjunctive treatment for preventing sepsis and/or multi-organ failure.

**Keywords:** activated carbon; activated charcoal; sepsis; inflammation; pathogen-associated molecular patterns; damage-associated molecular patterns

## 1. Introduction

Over 48.9 million people are affected by sepsis each year and mortality associated with sepsis and septic shock reaches 11 million annually varying with age, individual predisposing genetic influence, comorbidities, access to health care, and its provision [1]. The treatment of sepsis is essentially limited to cardiovascular support, antibiotics, and fluid therapy. Prevention, early recognition, and prompt treatment of sepsis are of paramount importance in improving patient outcomes [2,3]. The lack of significant success to find effective therapies is in large part due to the heterogeneity of factors determining the onset and development of sepsis (e.g., type of the causative organism, the route of entry, and individual host response) making sepsis research challenging. The current definition of sepsis is defined as "life-threatening organ dysfunction caused by a dysregulated host response to infection" [4]. The mechanism of this dysregulated response is partly dependent on pathogen-associated molecular patterns (PAMPs) that damage tissues and cells, triggering an abnormal cellular response to inflammation. This dysregulated cellular reaction to PAMPs leads, in turn, to overproduction and release (into biological fluids and tissues)

of the damage-associated molecular patterns (DAMPs) including cytokines, free radicals, C5a, membrane-attack molecules, HMGB-1 further enhancing tissue and organ injury [5]. The levels of these PAMPs and DAMPs are also abnormal in biological fluids of organisms with different pathologies and essentially reflect the inflammation state that accompanies them [6,7]. Normalization of PAMPs and DAMPs levels in these pathological states is often a sign of clinical amelioration and this normalization can be also the therapeutical target [8–11]. Activated carbon or activated charcoal (AC) is a well-known remedy and, today has enhanced biocompatibility and high-adsorptive capacity [12]. It can adsorb different inflammation-related PAMPs and DAMPs by non-selective (broad-range) [10,12,13], selective [14], and combined (non-selective and selective) [15–18] from biological fluids. AC is administered orally, per rectum, or via the hemoperfusion system for mitigation of different pathological states accompanied by inflammation, and has an impact on different parameters of homeostasis [19–21].

In the above context, this review posits several objectives:

– identification of the pathological states with an abnormal level of inflammation markers—PAMPs and DAMPs, in which AC administration is beneficial;
– identification of homeostasis parameters, compromised by inflammation and/or sepsis, which can be ameliorated by AC therapy;
– delineation of the known and potential mechanisms of action of AC therapy;
– identification of the challenges, and future areas/options of AC applications to maximize its efficiency.

## 2. First Application of AC for Infection Prevention and Treatment

The application of AC for preventing and treating infection has been well-known since ancient times. Phoenician trading ships preserved water in charred wooden barrels and AC was applied to treat wounds, poisoning, diarrhea, and flatulence already rife in Egypt [22] around 1500 B.C., by Hippocrates 400 B.C. [23] and by Claudius Galen [24]. Despite the modern controversy regarding activated charcoal's efficacy in the treatment of poisoning and diarrhea, it continues to be widely used. The existing controversy and contradiction in the AC efficacy are largely due to multiple confounding factors, e.g., the nature of poisoning and diarrhea, non-professional use, different toleration of AC in patients, and varying biocompatibility and adsorption capacities [12]. The emergence of new, more effective, and biocompatible AC compositions could improve their treatment efficacy [25]. Enterosorbent AC, with an ameliorated biocompatibility (uniform distribution in the intestinal content without formation of agglomerates) and more developed adsorptive surface (represented by not only micro- but especially by meso-, and macro-pores), has the potential to be more effective at much lower doses [12]. Adsorption tests relating to *Escherichia coli* on commercially available ACs have demonstrated that the sorption capacity of AC rose proportionally to the increasing volume of macropores and hydrophobicity of the surface [26].

## 3. AC for Uncontrolled Inflammation Treatment

### 3.1. AC Application for a Decrease PAMPs and DAMPs Elevated Levels

The progression of many diseases is due to an aberrant, dysregulated inflammation, often leading to sepsis and/or organ failure. Under normal circumstances, the immune system triggers an orchestrated defense process known as inflammation in response to infection/injury. Exogenous PAMPs (pathogen-associated molecular patterns), and endogenous DAMPs (damage-associated molecular patterns) are released into the blood/tissues and are recognized by the so-called Pattern Recognition Receptors (PRRs). Uncontrolled PRRs activation by highly elevated levels of PAMPs and DAMPs can lead to the development of aberrant inflammation phenotypes underlying various diseases such as autoimmune aggression, arthritis, cancer, and sepsis [5]. A therapeutic strategy aiming at decreasing elevated PAMPs and DAMPs levels in circulation appears justified [5,9,10,13,27]. Activated carbon can be considered as a candidate to regulate PAMPs and DAMPs levels by

selective or/and broad range adsorption of exogenous and endogenous inflammation and sepsis-related pathogens from biological fluids, including blood, plasma, and intestinal juice. There are numerous reports of beneficial AC applications in pre-clinical, clinical studies, and randomized clinical trials [20,28–32] to confirm AC efficacy and safety for inflammation mitigation and/or MODS prevention and treatment. Table 1 highlights some PAMPs and DAMPs that can be adsorbed by AC.

**Table 1.** List of some PAMPs & DAMPs adsorbed and adsorbed potentially *, ##.

| | Substances | Adsorbed, [Reference Number] | MW, kDa | Adsorbed Potentially * |
|---|---|---|---|---|
| PAMPs: | Toxic Shock Syndrome Toxin-1 (TSST-1) | - | 24 | * |
| | Streptococcal cysteine protease (SpeB) | - | 40 | * |
| | Bacteria and viruses | [33–35] | | |
| | Shiga toxin (Stx-2B) | [33] | | |
| | Clostridium difficile toxins: TcDA, and TcDB | [33] | | |
| | Cyanotoxins | [36] | | |
| | Endotoxin | [33,37] | | |
| | Mycotoxins: aflatoxin B1-312 Da and T-2 toxin | [38] | | |
| DAMPs: | Calprotectin | - | 36.5 | * |
| | PCT (procalcitonin) | - | 14.5 | * |
| | high-mobility group protein 1 (HMGB-1) | - | 25 | * |
| | S100-family protein | - | 9–12 | * |
| | Small heat shock proteins | [39] | | |
| | Histones | - | 11–22 | |
| | Myoglobin | [40] | | |
| | Azurocidin | - | 37 | * |
| | Plasminogen activator inhibitor1 (PAI-1) | - | 43 | * |
| Cytokines pro-inflammatory & anti-inflammatory | C3a | [13] | | |
| | C5a | [13,27] | | |
| | IFN- α, IFN-γ, TNF-α, IL-1β, IL-6 | [37] | | |
| | G-CSF | - | 18.8 | * |
| | GM-CSF | - | 14–35 | |
| | IL-9 | - | 14 | * |
| | IL-1 | [37] | | |
| | IL-2, IL-3, IL-8 | [13,41] | | |
| | IL-18 | - | 18.2 | * |
| | IL-4, IL-10 | [42] | | |
| | IL-1ra | - | 22–25 | * |
| | IL-13 | - | 12.5 | * |
| Drugs | Cytostatic | [43] | | |
| | Antibiotics | [29] | | |
| | Heparin | [44] | | |
| Chemokines | Eotaxin | - | 8.4 | * |
| | CXCL-1 | - | 11 | * |
| | MCP-1 | - | 11–13 | * |
| | MIP-1alpha | - | 7.8 | * |
| Hormone | T3 Cortisol | - | 0.362 | * |
| Metabolization product | Ammonia | [30,45] | | |
| | Creatinine | [17] | | |
| | Lactic acid | [46] | | |
| | Bile acids | [42] | | |
| Protein-bound substances | Unconjugated bilirubin, uremic toxin (CMPF), indoxyl sulfate, hippuric acid | [17] | | |
| | Indoles, scatoles, phenols, polyamines, advanced glycation end's product (AGES) | [47] | | |
| Cells/cellular component | Blood cells: platelets, neutrophils, monocytes | [48,49] | | |
| | Extracellular vesicles (EVs) | ## | | |

* can be adsorbed potentially accordingly, to their relatively low molecular weight (<50 KD) from biological fluids by an uncoated AC. ## Extracellular vesicles (EVs).

Potential benefits of EVs removal by adsorption remain unexplored due to the different EVs detection/identification techniques and the fact that EVs are only emerging biomarkers in the critical care field. EVs in sepsis (as many other biomarkers) can play dual, either detrimental or beneficial roles that likely depend on the disease phase, EVs' cellular source, site, and others [50]. The first clinical study [51] reports that the level of (MVs) microvesicles

(part of EVs) in the size range of 0.1–1 μm does not change in the blood of patients after hemoperfusion during cardiopulmonary bypass surgery compared to the control. MVs are, most likely, detached from some complexes after contact with the adsorbent leaving the MVs count largely unaffected. This possible mechanism is described in a recent in vitro study investigating the adsorbent's effect on free C-reactive protein (CRP) and CRP complexed with EVs in plasma from septic patients [52].

### 3.2. Adsorption and AC Modification

Despite the long history of AC application, the mechanism of the therapeutic action of AC is not completely clear. Yet, the physical adsorption of noxious substances present in biological fluids seems to constitute the main contributor [19]. The physical adsorption, conditioned by weak Van der Waals forces is reversible and depends among others on the AC porosity parameters, the nature of adsorbed substances, their concentration, and the condition of the process itself. This means that AC adsorbs components of the fluid's solutes at different rates independently of their role and function in the body. The adsorption rate or kinetics (an important index for medical AC design) of these substances is in direct relation to their molecular weight and concentration. It additionally depends on their chemical and structural parameters as well as on those of AC parameters. An enhancement of AC-dependent adsorption can be modified in several ways, e.g., (i) AC additional activation for surface functionalization and porosity development; (ii) coating; (iii) ligand grafting, and more. The key structural porosity factors that positively affect the adsorption rate are (a) inter-connectivity [53] and (b) fractality [54]. A high adsorption rate is especially important when it comes to removing toxins strongly bound to serum proteins such as albumin. Plasma concentrations of many uremic toxins (indoxyl sulfate, p-cresyl sulfate) [55] and hepatic toxin unconjugated bilirubin [56] are frequently elevated in sepsis. These toxins have high-association constants with serum albumin to form bulky complexes. Effective reduction of such complexes has become possible with the development of carbon adsorbents with tailored porosity features such as fractality [17,57,58]. AC coated [59] and/or grafted with different ligands [18,60] acquires new properties like adsorption selectivity and improved biocompatibility. AC becomes bi-functional and permits the simultaneous withdrawal of the target substances by coupling with the ligand and by a broad range (non-selective) adsorption of PAMPs and DAMPs (Table 1). For example, the combination of selective adsorption by such ligands as DNA or dextran sulfate fixed on AC with non-selective adsorption of activated charcoal (AC) ameliorates adsorption of DNA-binding substances and proinflammatory cytokines, compared to uncoated AC [18]. DNA-containing activated charcoal for direct hemoperfusion in a clinical trial of bronchial asthma treatment appeared to be more effective than the same uncoated AC [60]. The inhibitory properties of highly activated AC against the development of infection caused by herpes simplex virus were enhanced by the antiviral drug acyclovir (ACV) adsorbed on AC. Adsorbed ACV is apparently released from AC, only in the presence of the virus and AC adsorbing herpes simplex virus itself, and also delivers the ACV drug to the area of inflammation [34]. It was shown [61] that gentamycin and sulfamethoxazole adsorbed by commercial pharmaceutical activated carbon (in the absence of their desorption from AC) partly retain (25–50% reduction) their inhibitory effect against the growth of *E. coli* and *S. aureus*. Modified Ordered Mesoporous Carbon doped with cerium enables an enhancement of the resorcinol adsorption [62].

### 3.3. Per Oral Administration of Activated Charcoal (Enterosorption)

"The gut is open to the outer environment, harbors the microbiome containing several folds more genetic materials than the human genome and produces a myriad of metabolites as well as hormones/peptides" [63]. This clearly explains why the gut-kidney/liver/brain cross-talks in the sepsis-induced injury of those organs are an important area of research to the scientific community [63–65]. Those gut-organ interaction pathways in various critical

illness states create an opening for potential therapeutic interventions via the oral AC application.

### 3.3.1. AC in Anti-Diarrhea Treatment

Diarrhea is associated with severe sepsis in adult patients with ileus, acute kidney injury, metabolic acidosis, hypocalcemia, and those receiving steroids. Gram-negative bacteria are predominant in diarrheal adults who have severe sepsis [66]. Bile acids are the natural antidotes to *E. coli* endotoxin [67] and AC can adsorb bile acid [42]. A high concentration of bile acids in the colon is often a cause of chronic diarrhea and some patients with diarrhea-predominant irritable bowel syndrome have associated bile acid malabsorption [68] (currently treated by bile acid sequestrants). It was shown that the enterosorption with AC in the mouse model decreased the concentration of proinflammatory mediators, not only in the intestinal juice but also in the liver and kidney [69]. Systemic immunoprotection and correction of homeostasis [70,71], and inhibition of viral replication [71], are possible mechanisms of AC action in the diarrhea treatment.

### 3.3.2. AC Enterosorption in Renal Dysfunction

Sepsis-associated acute kidney injury (S-AKI) is a frequent complication in critically ill patients [72,73]. Chronic kidney disease (CKD) progression in patients is marked by the serum retention of toxic compounds, resulting in uremic syndrome [74]. The strategy of uremic toxins removal by adsorption during CKD is discussed in the reviews [75,76].

Uremic Toxins

Uremic toxins are categorized as small water-soluble molecules, protein-bound compounds, and middle molecules [77]. The most hazardous are those that are difficult to remove by dialysis: protein-bound uremic toxins (PBUTs). Many PBUTs originate from the gut and they can be separated into two groups: (1) diet-ingested toxins and (2) toxins generated by the gut microbial metabolism. These PBUTs are represented by advanced glycation end-products, hippurates, indoles, phenols, polyamines, etc. PBUTs, regardless of the group, impact the gut microflora composition, and their levels in serum are increased with the progression of kidney disease. Gut dysbiosis causes an overproduction of uremic toxins, negatively affecting intestinal permeability and the immune system [78].

Mechanistic Aspects of Enterosorption Action in Renal Dysfunction

The concentration of uremic toxins can be reduced by the administration of adsorbents with a high-adsorptive capacity and mesoporosity, especially for PBUT's removal [17,79,80]. In vitro, hemoperfusion experiments [58] demonstrated that AC is capable of adsorbing several PBUTs, such as indoxyl sulfate, p-cresyl sulfate, indole acetic acid, phenyl sulfate, and hippuric acid from the blood. Per oral administration of AC reduces the cumulation of indoxyl sulfate (IS) and p-cresyl sulfate (p-CS)—the major PBUTs not eliminated by dialysis in the blood of patients with CKD [81]. The mechanism of this uremic toxin's reduction in the blood is related to the adsorption from the gut of PBUTs initiators, such as aromatic amino acids tryptophan (producer of indole -the precursor of IS), phenylalanine, tyrosine, and the precursors of IS and PCS—indole and p-cresol [47,81]. AC administration can reduce systemic oxidative stress and inflammation [81,82]. A clinical investigation of orally administered AC in hemodialysis patients demonstrated a significant decrease of oxidized albumin and 8-isoprostane—the pathogens and biomarkers of lipid peroxidation in the plasma [83]. Another possible mechanism of uremic toxins reduction is the amelioration of intestinal barrier disruption, which contributes to the blockade of the absorption of these toxic molecules. It is argued that systemic, chronic, and uncontrolled inflammation can be evoked by the intrusion into the blood of various toxins from a leaky gut [84]. Several studies demonstrated that AC enterosorption can restore the gut barrier function, thereby reducing toxin leakage [19,32,85]. An oral application of AC in rats reversed the CKD-induced disruption of the colonic epithelial tight junctions, as well as the associated

endotoxemia, oxidative stress, and inflammation [19,32]. This restoration of the gut barrier function by AC was related to the microbiota improvement expressed in the increase of the Lactobacillus and Bacteroides populations [19]. Animal studies revealed that AC may modulate the gut environment and microbiota composition [19,86,87]. Sato et al. [86] have reported that at least 23 gut microbes were significantly changed by renal failure after AC oral administration and changes of *Erysipelotrichaceae*, *Clostridium sensu* stricto 1, *Roseburia*, *Faecalibaculum*, *Blautia* and *Desulfovibrio* were correlated with amelioration of fecal p-cresol production. This intestinal modulation may in part explain the different attenuation effects of AST-120 on serum IS and pCS concentrations noted in various studies [47]. A study of the microbiome in CKD patients revealed that significant gut dysbiosis changes are partially restored by enterosorption with AC. AC intervenes in a possible signature of short- and medium-chain fatty acid's metabolism [88]. AC per oral administration was reported in multiple investigations to decrease the level of PBUTs in the plasma of CKD patients [47,76,81,89], etc., and evidently contributes to the restoration of structure and function of serum albumin, as suggested in [90]. A 2019 clinical trial reported that AC, with a surface of 1000–1400 m$^2$ per/g, effectively delayed the onset of hyperphosphatemia, inhibiting the development of vascular and valve calcification in non-dialysis patients with CKD (stage 3–4) [91]. The mechanism of AC action on phosphate levels is unclear but can be related to direct gut adsorption of some inflammatory biomarkers detected in feces. Such fecal biomarkers as calprotectin, elastase, and lactoferrin are derived from neutrophils, activated in response to chronic inflammation and infiltrating gut mucosa [92]. Their elevated levels are typically indicative of inflammatory bowel diseases [93] and are related to severe acute respiratory syndrome coronavirus 2 (SARS-CoV-2) and to Metabolic-associated fatty liver disease [94]. Circulating calprotectin in dialysis patients with CKD is two times higher than in healthy subjects and it strongly correlates with the circulating phosphate [95]. Moreover, an elevated plasma calprotectin was associated with all-cause mortality in the general population [96]. In CKD patients, this association was observed in the high-phosphate group of patients, but not among the low-phosphate group [95]. The increased circulating free active elastase released from the primed peripheral polymorphonuclear leukocytes, together with the higher cell surface-bound enzymes and the lower levels of $\alpha$1-AT (the natural elastase inhibitor antitrypsin), resulted in a higher elastase activity in sera of patients with dialysis [97]. This exacerbated elastase activity could lead to endothelial dysfunction and subsequent atherosclerosis and cardiovascular complications common in CKD patients. Both calprotectin (molecular weight of 36.5 kDa) and elastase (29.5 kDa) can be effectively adsorbed by AC from the gut, thus reducing their accumulation and thereby slowing/preventing the development of the vascular and valve calcification in CKD patients.

Clinical Interventions

Long-term clinical trials of AC in progressive renal failure were conducted with the spherical carbon adsorbent AST-120 (Kremezin), and were approved in Japan in 1991. Large randomized, placebo-controlled trials (EPPIC-1 and EPPIC-2) [98] found that AST-120 did not have a significant effect on different endpoints, including dialysis initiation, kidney transplantation, and serum creatinine doubling in patients with Stage-4 CKD, despite a shorter estimated median time to primary endpoints for the placebo group (124 vs. 170–189 months in the AST-120 groups) [47]. But in patients with Stage-5 CKD, AC demonstrated a significant improvement of uremic symptoms [28,82]. Cupisti et al. [47] underline that the beneficial effect of AC administration is more pronounced in groups of patients with severe CKD or with higher comorbidity. In patients with diabetic nephropathy, a post hoc analysis of the EPPIC trial suggested a positive effect on patients with relevant proteinuria (>0.5 g/g of creatinine) [98,99]. Clinical study [89] showed that enterosorbent AC improves flow-mediated, endothelium-dependent vasodilatation and the oxidized glutathione/glutathione ratio (indicative of the oxidative status of tissues and body fluids) [100].

### 3.3.3. AC Enterosorption in Liver Failure

The liver has a central role in sepsis pathophysiology given that it regulates immune defense during systemic infections [56]. However, the liver is also a prime target for sepsis-related injury and hepatic dysfunction substantially impairs the prognosis of sepsis in the intensive care unit [101]. The liver-gut axis has been demonstrated to contribute to the pathogenesis of most liver diseases in experimental models and clinical investigations [101]. Gut microbiome changes, combined with increased intestinal permeability, induce an accumulation of gut-derived metabolites and bacteria in the liver. Infiltration of the liver by responsive immune cells often leads to chronic liver inflammation, cirrhosis, and organ failure [102]. The elevated counts of Bacteroides populations (Peptidostreptococcaceae, Clostridium XI) in the stool of bile-duct ligated cirrhotic animals significantly decreased after AC oral administration. The AC significantly lowers the levels of citrate, dimethylarginine, and creatinine. AC treatment increases creatinine and bile acid levels and decreases glycine levels in urine. AC inhibits the expression of IL-18/IL-1ß and reduces LPS-induced monocyte ROS production [103]. Yet another study showed that AC significantly decreased the portal blood pressure, circulating ALT, and LPS-induced ROS production by the Kupfer cells; a body weight increase in AC-treated cirrhotic rats was also observed [42]. In the rats subjected to bile duct ligation, the RNA-seq analysis shows that AC (Yaq-001) attenuates the inflammation in the ileum. This AC treatment contributes to the activation of the defense system by regulating genes in the liver involved in the inhibition of oxidative stress [104]. In a model of leptin-null mice with non-alcoholic steatohepatitis (NASH) AC treatment was associated with a significant reduction of circulating ALT and ROS production by Kupfer cells [105]. Hepatic TLR-4 expression is inhibited in carbon-treated mice compared to non-treated controls. A significant reduction in the $F4/80^+$, $CD68^-$, $CD11b^+$ cell subpopulation is found as the effect of oral-carbon treatment [105]. And the action of this hepatic $F4/80^+$, $CD11b^+$, and $CD68^-$ cells plays a key role in the antibacterial response against gut-associated sepsis [106]. An experimental study with nonalcoholic fatty liver disease (NAFLD) in mice suggested that AC (AST-120) treatment alters the fecal microbiota composition and significantly decreases hepatic weight and hepatic triglyceride levels [87].

### Clinical Study

Oral AC—Carbalive (Yaq-001, synthetic activated carbon) administration to patients with diuretic-responsive cirrhotic ascites demonstrated a reduction of circulating white blood cell (WBC) count, C-reactive protein (CRP), IL-6 and C-X-C motif chemokine ligand 10 (CXCL10) [30]. AC-Carbalive also improved the fecal cytokine concentration (IL-17A, TNF-$\alpha$), gut permeability (measured by the decrease of plasma D-lactate), and regulated metabolism (reduction of stool ammonia, and increase in the urinary hippurate) [30].

### 3.4. AC Application per Rectum

The rectal AC application is not common. However, the discovery of microbiota's association with the pathogenesis of many inflammatory diseases and infections [107], concurrent with a robust presence of inflammation biomarkers in the feces [92], is likely to resurrect this administration route.

DAMPs in inflammatory bowel diseases are considered therapeutic targets and not solely biomarkers indicative of intestinal inflammation [11]. AC, when introduced in the most abundant microbiota part of the gut—the large intestine can adsorb the part of inflammatory pathogens PAMPs and DAMPs [11,92], and free antibiotics without affecting their levels in blood [29]. AC was shown to exert a positive action on microbiota [19,29,108,109]. Zawadzki et al. [20] tested a rectal application of AC (8–24 week duration) in patients with chronic, uncomplicated perianal fistulas who were scheduled for surgery. The etiology of perianal fistulas is believed to be cryptoglandular infection, often in individuals with a previous history of an anorectal abscess, malignancy, Crohn's disease, and radiation therapy. The recurrence rate of AC-treated patients in that trial was 75%—the same rate as for the fistula plug treatment. Notably, the results of the oral AC administration in the treatment

of anal fistula in Crohn's disease are conflicting; the authors suggested that AC passage throughout the intestinal tract reduces the adsorptive capacity of the adsorbent [110,111]. While AC reports on using the rectal route are scarce, the existing evidence and superior access to the gut microbiome intuitively present this approach as a prospective and safe method of inflammation mitigation from various origins.

*3.5. Hemoperfusion (HP)*

Hemoperfusion is a procedure of blood purification by non-selective or selective adsorption of different plasma solutes by granules (beads) of AC or (polymeric adsorbent). A recent review of Ronco C & Bellomo R [112] addresses a contradictory history of hemoperfusion as well as its state of the art. The results of clinical trials of selective HP targeting key molecules, e.g., endotoxin (Toramyxin™ cartridge) and of non-selective HP (adsorption of the broad range of PAMPs &DAMPs molecules-CytoSorb® device, JAFRON HA cartridge) in sepsis treatment are discussed. HP in the clinic for sepsis and critical COVID-19 treatment was applied primarily for the removal of different inflammatory mediators such as cytokines and endotoxin [21].

Both selective and non-selective HP procedures are reported to have a beneficial impact on dysregulated inflammatory conditions such as sepsis, COVID-19, and influenza, but definitive therapeutic efficacy is not yet confirmed in clinical trials [9,113,114].

3.5.1. The Strategy of Membrane-Damage Substance Removal in Sepsis

Different PAMPs and DAMPs (Table 1) can be removed by selective HP [14] targeting key molecules or non-selective HP by porous materials [27]. The part of PAMPs and DAMPs with the molecular weight range, of 5–60 kDa adsorbed with Hp device in sepsis and septic shock treatment, is classified as membrane-damaged substances (Table 2). There, the authors [115] suggest that the damage of cell membranes by extracellularly elevated levels of phospholipases (PLA2), RONS (reactive oxygen and nitrogen species), pore-forming proteins (PFPs), alongside the dysregulation of osmotic homeostasis in response to inflammation, is one of the main mechanisms of multiple organ failure (MOF), that often occurs in sepsis. Mechanisms causing MOF are still unclear but they must be reversible, at least at the beginning of the disease, and should induce cell dysfunction, but not cell death [116,117]. Damaged cytoplasmic membranes can be efficiently repaired and the rate of their recovery depends on the level of the lesion and the repair capacity of the organism. Alternatively, it can be transformed into a non-resolving phase if repair mechanisms are not sufficient or if the damage is extended to intracellular compartments essential for vital cellular functions [115]. Possibly, that is why the evident statistical improvement of hemodynamics (the critical factor in initiating sepsis) was found only in the case of hemoperfusion application in the early stage of sepsis [114]. The early application of HA 330 hemoperfusion (within 72 h of sepsis diagnosis) in patients with sepsis or septic shock is reported to be beneficial in comparison with late HP procedures > 72 h and is recommended for randomized study [118].

**Table 2.** Some cell membrane damaging factors with an elevated level in serum or plasma: phospholipases (PLA), pore-forming proteins (PFTs or PFPs)-bacterial toxins, and pore-forming membrane attack complex of complement (MAC).Molecular weight-MW, isoelectric point for proteins -pI.

| Substances Upregulated in Serum or Plasma | MW (kDa) | pI | Reference |
|---|---|---|---|
| Secretory phospholipase A2 (sPLA 2) group IIa | 13–17 | 6.68 | [119] |
| cytosolic PLA 2 (cPLA2) | 80–85 group IV | 5.0 | [120] |
| C5 (MAC) | $120(\alpha) + 175(\beta) = 195$ | 4.7–5.5 | [121] |
| C5a (MAC) | 11 | 8.6 | [121] |
| C5b (MAC) | 171 | 4.5–5.3 | [121] |
| C6 (MAC) | 90–100 | 5.6–6.1 | [122] |

**Table 2.** *Cont.*

| Substances Upregulated in Serum or Plasma | MW (kDa) | pI | Reference |
|---|---|---|---|
| C7(MAC) | 110 | 5.6–6.1 | [122] |
| C8 (MAC) | 64 ($\alpha$) + 64($\beta$) + 22($\gamma$) = 150 | 7.4–7.9 | [123] |
| Lysenin PFTs | 41 | 6.5 | [124] |
| Alpha-hemolysin ($\alpha$HL) secreted from Staphylococcus aureus PFTs | 33–37 | 7.94 | [125–127] |
| $\gamma$-Hemolysin, from *Staphylococcus aureus* PFTs | 31.81 | 9.52 | [125,128] |
| Delta-hemolysin from *Staphylococcus aureus* | 3 | - | [127] |
| Beta toxin from Clostridium perfringens (CPB) $\beta$-PFTs | 27.6 | 5.6–6 | [129] |
| Leucocidin A/B (LukAB) *from Staphylococcus aureus* PFTs | 32.39 | 8.77 | [125,127,130] |
| Aerolysin PFTs | 50–53 | 5–7 | [131,132] |
| Histone H4 | 11 | 8–9 | [133–135] |

### 3.5.2. Combination of Selective and Non-Selective Adsorption for HP

The combination of broad-range adsorption of PAMPs and DAMPs with selective ones seems to be an interesting approach for application in sepsis. The range of membrane-attacking substances, especially from staphylococcal strains and histones show similarities with isoelectric points of 7.3–9.5 and are basic proteins (Table 2).

Dramatically elevated levels of extracellular basic protein-histones in plasma, predict the mortality of patients with sepsis [133,136]. These histones are particularly interesting as membrane-damaging agents [134,137]. The serum level of histone H3 is three-fold higher in patients with MODS than those in patients with infectious diseases without organ failure [136]. Histones act not only as deregulators of osmotic homeostasis through cell permeabilization and calcium influx [138] but can act directly, as histone H4 on cell membranes without necessarily binding to known cell membrane receptors [134]. H4 directly stimulates neutrophil activation through membrane permeabilization [134]. Extracellular histone contributes to the dysfunction of human endothelial cells by their activation for the production of pro-inflammatory cytokines and increases adhesion molecule expression [139].

Membrane-attacking substances from Staphylococcal strains and histones mainly act through electrostatic interactions with negatively charged phospholipids of the membrane, the acidic domains of receptors, and can be removed better with negatively charged adsorbents, e.g., coated with heparin [16,44], polysialic [140], lactic acids [141]. Particularly interesting is the heparin-coated adsorbent. Non-selective broad-range HP can obtain some selectivity due to the adsorption from the blood of the anticoagulant heparin on porous adsorbent as activated charcoal [44]. A recent study demonstrates that immobilized heparin can remove effectors of deregulated immunothrombosis, such as activated platelets and platelet-derived extracellular vesicles (pEVs) expressing PF4, soluble PF4, HMGB1, histones, as well as histone-decorated NETs. Authors suggest the usefulness of heparin-containing adsorbent application for the mitigation of sepsis thrombotic complications [16]. Heparin-binding cationic protein, antimicrobial azurocidin [142], was found [143] to express elevated levels in the plasma of patients with sepsis several hours before they develop hypotension or organ dysfunction. The decrease in this protein level can probably prevent multiple organ failure.

The binary adsorbent or association of both HP systems, combining non-selective with selective removal of PAMPs and DAMPs, is a prospective approach for HP device development for sepsis treatment.

An early application of non-selective, selective, or combined (non-selective and selective) hemoperfusion, before irreparable membrane damage and cell destruction for the prevention of multiple organ failure and septic shock, is a prospective therapeutical approach.

**4. Positive Effect of AC on Some Parameters of Homeostasis Compromised by Sepsis**

*4.1. Bone Marrow Index and Oxidative Stress*

Sepsis is a dramatic example of an inadequate host bone marrow (BM) response to infection, whereby an initial neutrophilia and hyper-reactive immune response is followed by profound neutropenia, leukocyte hypo-responsiveness, and an inability of the host to control the bacterial attack [144]. Leukopenia is associated with a higher risk of mortality than leukocytosis among patients with Systemic Inflammatory Response Syndrome [145]. The potential role of inflammatory mediators in the dysregulation of bone marrow cell function during abdominal sepsis is discussed by Barthlen and coauthors [146]. Oxidative stress, age-related inflammation ("inflammaging"), and cellular senescence appear to play a major role in BM inflammation, impairing the long-term maintenance of immunological memory [147]. The role of oxidative stress in the development of sepsis, and disease severity was demonstrated in numerous publications [148,149].

Enterosorption with highly activated AC with S BET 2200–2500 $m^2$ per/g helps to reduce oxidative stress and, possibly, by this way, accelerates the bone marrow restoration in experiments with rats after high-dose injections of the anti-cancer drug (melphalan), and after total body X-irradiation [150–152]. Monocyte LPS-induced ROS was attenuated with oral AC treatment in cirrhotic rats [103]. AC reduces the superoxide radical generation rate (SOR) in the liver and brain [150], attenuates manifestations of oxidative stress [150–153], and decreases nitric oxide levels in lymph nodes in experimental animal models with severe inflammation [69].

*4.2. Serum Albumin*

COVID-19-induced oxidative stress inflicts structural damage to human serum albumin (HSA) and is linked with mortality outcomes in critically ill patients. And the transport function of HSA, as well as its physical and chemical properties in non-survivors, are dramatically changed compared to that of survivors and healthy patients [154]. Recovery of conformation of native albumin molecule in blood plasma [153] accompanied by the mitigation of oxidative stress in rats with severe intoxication caused by CCl4 injection, confirms high-detoxification properties of biocompatible, highly activated AC enterosorbents. Clinical investigations of patients with renal failure [83,89] receiving AC via oral administration demonstrate the mitigation of oxidative stress and reduction of pathological forms of albumin [83]. A recent publication reveals another interesting way of albumin conformation restoration with the help of carbon particles (CPs) derived from AC. These CPs have a low-binding affinity to albumin and purify albumin molecules from the trace concentrations of strongly bound ligands [155].

*4.3. Microbiome*

A growing interest in the role of gut functionality/interaction in critical illnesses and pathological states, such as sepsis and MODS, put the microbiome in the limelight. The microbiome in sepsis is profoundly distorted, including a sharp decrease in diversity, the loss of commensal bacteria, and the overgrowth of potentially pathogenic bacteria [156]. It was found that the microbiota in sepsis patients is rich in microorganisms such as *Parabacteroides*, *Fusobacterium*, and *Bilophylloma*, which are highly related to inflammation. An abundance of enterococcus species is increased in septic non-survivors [157]. Tissue bacteria and their metabolites reflect and/or can directly contribute to the development of metabolic diseases [158]. The review of Kullberg et al [159] discusses the effects and risks of some microbiota-targeted preventive and therapeutic strategies in sepsis.

In multiple animal experiments and clinical studies, it was shown that AC administered orally contributes to the amelioration of the gut microbiota composition affected in different pathological states [19,29,86,87]. This can be explained by broad-range adsorption by AC of some parts of gut-derived PAMPs and DAMPs such as microbe-associated molecular patterns, including bacteria [33,160], fecal cytokines [30], ammonia [30,45], endotoxin [33], and also the products from the interaction of the microbiota with the dietary

substances such as acetaldehyde [161], trimethylamine [162], ethanol [163], and free fatty acids [164].

In addition, blood-derived PAMPs and DAMPs (Table 1) from 7–8 L of blood filtrate that enters daily in the gastrointestinal tract, comprising mucus-1.5 L, gastric juice-1.5–2 L, bile-0.5–0.6 L, intestinal juice-3 L, can be adsorbed.

Some adsorptive "selectivity" of pathogenic microorganisms by AC can be another mechanism of gut microbiota regulation. In the colon, as part of the gut with the highest abundance of microbiota, AC will adsorb firstly more accessible pathogenic bacteria that are out of cecal and proximal colon crypts colonized by the commensal microflora [165], which defends epithelial cell homeostasis [166]. Pathogenic microorganisms, concentrated near damaged intestinal mucus with decreased cross-linking density [167], must be preferentially adsorbed by AC. Biocompatible AC (especially in the form of carbon fiber chopping) has a positive impact on intestinal contents. AC changes the texture, properties, and volume of undigested residue, helping to maintain and modify the diet and acting as non-digestible dietary fibers [168]. A recent clinical trial [169] performed on healthy volunteers, receiving only powdered USP AC with orange juice for 2 weeks, demonstrated excellent safety and tolerability of AC without any unfavorable impact on the gut microbiota. Exposure to antibiotics affects the richness and composition of intestinal microbiota, which can have a negative impact on host health [29,109,170]. Changes in functional attributes of the microbiota, reduction of microbial diversity, formation, and selection of antibiotic-resistant strains make hosts more susceptible to infection with pathogens such as *Clostridioides difficile* [170]. A specially formulated adsorbent, DAV132 (administered per os and acting in the ileocecal region of the gut), protected intestinal microbiota in human volunteers and decreased the concentration of free antibiotics in feces. This adsorbent did not affect the antibiotic's level in the blood but substantially ameliorated the microbiota [29].

## 5. Discussion

The inflammatory states described in this review have multiple commonalities with sepsis and MODS. Among them are: the abnormal levels of PAMPs and DAMPs; hematological, immunological, and microbiome disorders; etc. Sepsis is typically associated with dramatic changes in homeostasis, regarding the immune-inflammatory responses, and acts simultaneously across multiple systems in the host. The organs most frequently affected are the kidney, liver, lung, central nervous system, hematologic system, and heart [171]. MODS is the hallmark of sepsis and central to the Sepsis-3 definition. The review provides a summary of AC administration via various application routes for the treatment of kidney and liver failures, diarrhea, COVID-19, sepsis, etc., and demonstrates the therapeutical efficacy of AC. This curative efficacy confirms the usefulness of AC application for sepsis prevention.

The mechanism of therapeutic action of activated carbons to mitigate inflammation is mainly based on a broad-range, non-selective or selective adsorption of abnormally elevated PAMPs and DAMPs (Table 1). The positive effects of AC administration occur despite an evident removal by AC of part of beneficial substances or substances with a dual role (anti-inflammatory regulator, chemokines, etc.). AC, unlike injected antibodies and antibiotics (with a high volume of distribution) that speedily block the main part of targets, can operate only with part of solutes distributed in biological fluids. It makes AC application more advantageous in a schema of uncontrolled inflammation prevention, reducing the toxic burden and supporting the habitual defensive mechanisms restoring homeostasis.

Regulation of PAMPS and DAMPs abnormal levels can be facilitated by the application of selective and/or broad-range adsorbents as activated charcoals. Broad-range non-selective adsorption of solutes can be combined with selective targets for the enhancement of AC removal efficacy. This approach is also supported by the fact that discoveries and classifications permanently supplement the list of PAMPs and DAMPs dysregulated in body fluids during inflammation evolution (Table 1). AC can act also as a carrier of adsorbed solutes, delivering them, e.g., antiviral drug-acyclovir [34], or protecting their effectiveness

and activity as antibiotics [61] and heparin [16,44]. Another way of enhancing the AC-based removal of noxious substances from biological fluids can be attended by amelioration of the kinetic of their adsorption. The high rate of adsorption is important, especially in the case of protein-bound toxin (e.g., indol, bilirubin) removal. The increase of AC's total adsorptive surface to 2000–2700 m$^2$/g and the development of meso- and macro-pores are important for the high-adsorption kinetic of noxious substances [17,79,172]. Perspective adsorbents for sepsis prevention are carbon nanoparticles (CNP)—new carbon material [173,174]. The CNP possesses intrinsic anti-inflammatory properties reducing the production of proinflammatory cytokines and modulating immune cell maturation [175]. Unfortunately, the bioaccumulation-related toxicity of CNP hinders its therapeutical potential in the treatment of sepsis and other inflammatory conditions.

Microbiota restoration and protection, reduction of oxidative stress, protection and acceleration of bone barrow restoration, and recovery of plasmatic albumin structure and function by AC application are the facts, encouraging the application of AC for sepsis prevention [19,29,83,87,89,150–153].

The strategy of early AC application per os, per rectum, or in the hemoperfusion system to prevent the development of MODS in sepsis (and other inflammatory conditions) is important for AC curative efficacy. From this point of view, AC administration's oral and rectal routes are more accessible and technically simpler. They permit a fast introduction of AC in the schema of uncontrolled inflammation treatment for MODS prevention. To meaningfully contribute to homeostasis restoration (Figure 1), AC must be administered early to prevent/reduce irreparable damage to cells by critically dysregulated PAMPs and DAMPs. Preventive therapy of uncontrolled chronic inflammation leading to sepsis and MODS, as stated in [176], should be combined with the administration of antibiotics, anti-inflammatory, and anti-oxidant drugs.

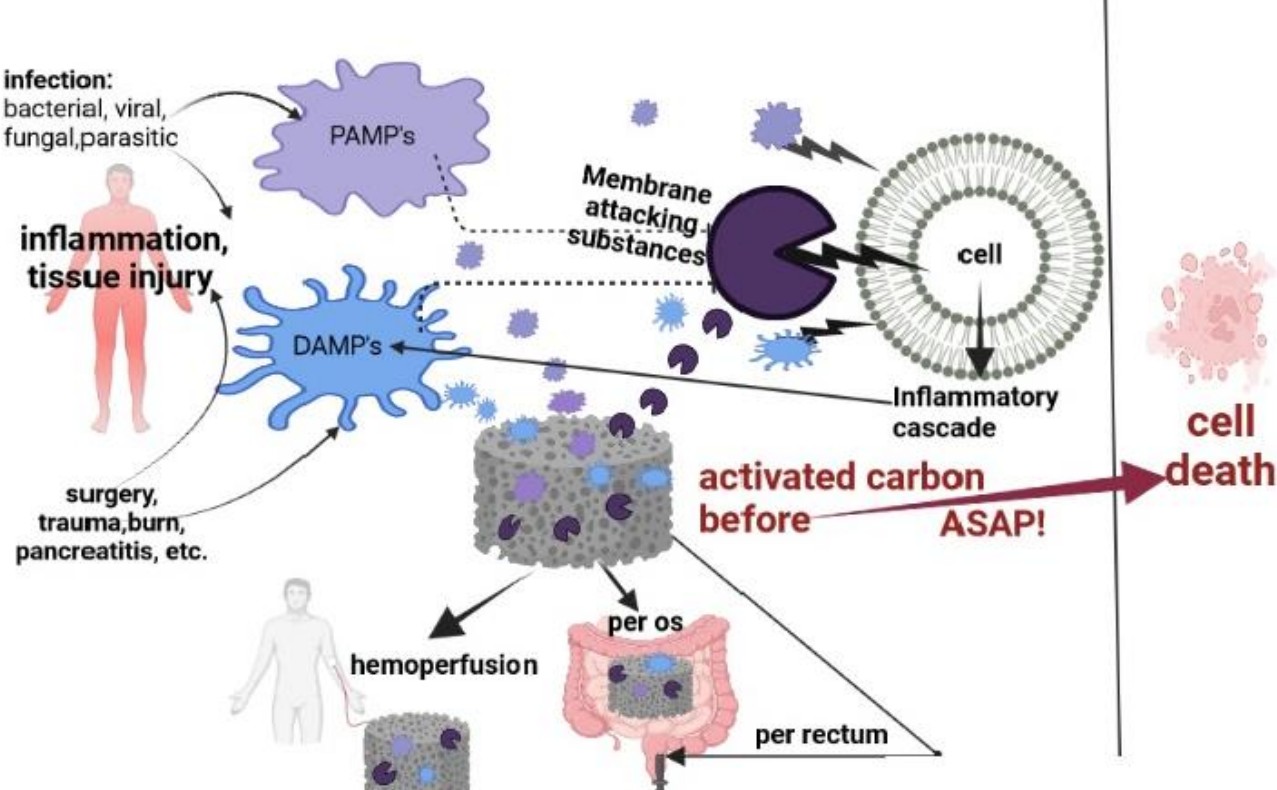

**Figure 1.** Administration of activated carbon orally, per rectum, or in the hemoperfusion system ASAP (as soon as possible) for the reduction of elevated levels of Pathogen-(PAMPs) and Damage-(DAMPs) Associated Patterns can protect cells/organs from death.

## 6. Conclusions

Table 3 demonstrates the achievements, shortcomings, and possible ways to cover the gaps in the strategy of AC development and application for the treatment of inflammation states, and for the correction of compromised parameters and systems. To summarize, biocompatible activated carbon with a high-adsorption capacity towards noxious substances, administered as soon as possible, protecting the microbiome and bone marrow, acting as an anti-inflammatory and anti-oxidant remedy constituting recognition as a promising candidate for sepsis prevention and intervention.

**Table 3.** Positive effects of AC administration via various application routes (per oral, per rectum, in hemoperfusion system) on chronic and acute inflammation states: diarrhea, renal dysfunction, liver failure, perianal fistula, sepsis, and on some parameters & systems compromised by these inflammations: PAMPs and DAMPs level, bone marrow, oxidative stress, serum albumin, microbiome. Proven/hypothesized mechanism of action(s) and gaps.

| Positive Effects of Activated Charcoal (AC) Administration on: | Proven/Hypothesized Mechanism of Action(s) | Gaps |
|---|---|---|
| PAMPs & DAMPs in biological fluids | Increasing AC surface & porosity and its selectivity towards PAMPs & DAMPs; by a combination of selective and broad-range AC adsorption; by modifying biological fluids e.g.,by transfusion, and special drug use to enhance PAMPs & DAMPs adsorption by AC. | a; adsorption of benefic substances |
| Bone marrow and oxidative stress | Early initiation of AC administration; study of the mechanisms of AC action. | b |
| Serum albumin | Increasing AC surface & porosity and selectivity towards protein-bound toxins. | a |
| Microbiome | Combination of different routes of AC administration; AC modification; study of mechanisms of AC action. | b; e |
| Diarrhea | Enhance AC biocompatibility and adsorptive capacity towards e.g., bacteria, viruses, and their toxins; study of the mechanisms of AC action. | a–d |
| Renal dysfunction | Early initiation of AC administration; combination of different routes of AC administration; amelioration of AC biocompatibility and adsorptive capacity towards uremic especially protein-bound toxins; study of the mechanisms of AC action. | b; d; e |
| Liver failure | Early initiation of AC administration; combination of different ways of AC administration; amelioration of AC biocompatibility and adsorptive capacity towards hepatic toxins; study of the mechanisms of AC action. | b; d; e |
| Perianal fistula | Combination of AC per oral administration with those per rectum; study of the mechanisms of AC action. | Begin of clinical application; b |
| Sepsis | Early initiation of AC administration before an irreparable membrane; a combination of different routes of AC administration; development of biocompatible AC combining selective and broad-range adsorption; study of the mechanisms of AC action. | a–d |

a—Low kinetic of adsorption of toxins and protein-bound toxins from biological fluids. b—Unclear mechanism of activated carbon (charcoal) action. c—Lack of sufficient biocompatibility of AC. d—Non-achieved clinical trials. e—Non-sufficient efficacy.

According to the IUPAC (International Union of Pure and Applied Chemistry), three groups of pores are distinguished according to the pore size: Macropores (>50 nm diameter); Mesopores (2–50 nm diameter); Micropores (<2 nm diameter).

**Author Contributions:** Conceptualization: H.R., M.O. and J.G.; writing—original draft preparation: E.S.; writing—review and editing: E.S., M.O. and H.R.; supervision: J.G. All authors have read and agreed to the published version of the manuscript.

**Funding:** This work was partly funded by FWF Austrian Science Fund (www.fwf.ac.at), grant number I5454 Ceus to MFO, and by AUVA.

**Data Availability Statement:** Not applicable.

**Acknowledgments:** Andrey Kozlov for interesting suggestions, Oleksandr Kozynchenko, Larysa Saknno for helpful discussions and advice.

**Conflicts of Interest:** The authors declare no conflict of interest.

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
