# Peer review of "Activated Carbon for Sepsis Prevention and Intervention: A Modern Way of Utilizing Old Therapies"

_carbon_

Round 1
Reviewer 1 Report
Comments-
The systematic review about Activated carbon for sepsis prevention is an interesting work and the authors have collected unique information from various electronic sources. The review is generally well written and impressive. However, in my opinion it has some minor shortcomings in regards to some data collections and text, Below I have provided numerous remarks on the manuscript which should be addressed-
1. In the last section of Introduction, clearly state that why this review is needed.
2. The aim of a review cannot be to summarize data, since they are already out in the public domain. You must develop a critical appraisal of the state of the art. This is an essential element of any review. There are important scientific questions (both conceptual and methodological) which need to be addressed with the primary studies. A review must highlight this. This is relevant for the main body of your text, your conclusions and the abstract.
3. Include the timeline in abstract until when the information is gathered.
4. Rewrite the keywords. Abbreviations should be avoided in keywords
5. The introduction should provide the relevant background and leads to well-defined objectives. Do not report results in the introduction or make conclusions in this section.
6. The ancient literatures, books, mythological texts etc. should also be used for collecting the information about modern way of utilizing old therapies from activated carbon.
7. In general, the language needs some minor revisions.
8. Overall we expect a critical assessment of the state of the art including precise and critical assessment of the papers reviewed (incl. concepts and methods).
9. Conclusions need to be critical and specific. It needs to highlight the achievements and specific scientific gaps in our knowledge. So what further research should have priority?
10. Lot of corrections are required in the reference. Carefully revise it.
Minor revisions on English. Carefully check the spellings and punctuations in the sentences.
Author Response
Dear Reviewer!
Thank you for your work and constructive remarks. Please find below our answers to your remarks:
- In the last section of the Introduction, clearly state that why this review is needed.
Answer: Thank you for this sober remark. We corrected the Introduction according to your recommendation.
- The aim of a review cannot be to summarize data, since they are already out in the public domain. You must develop a critical appraisal of the state of the art. This is an essential element of any review. There are important scientific questions (both conceptual and methodological) that need to be addressed with the primary studies. A review must highlight this. This is relevant for the main body of your text, your conclusions, and the abstract.
Answer: Thank you for this remark. We have now highlighted the objectives of our review in the Introduction (lines 71 – 78).
- Include the timeline in the abstract until when the information is gathered.
Answer: timeline is included in the abstract
- Rewrite the keywords. Abbreviations should be avoided in keywords
Answer: Keywords are rewritten
- The introduction should provide the relevant background and leads to well-defined objectives. Do not report results in the introduction or make conclusions in this section.
Answer: Thank you for this remark. We corrected the Introduction according to your recommendation.
- The ancient literatures, books, mythological texts etc. should also be used for collecting the information about modern way of utilizing old therapies from activated carbon.
Answer: True, we added them to the reference list.
- Overall we expect a critical assessment of the state of the art including precise and critical assessment of the papers reviewed (incl. concepts and methods).
Answer: Yes, we attempted to do that. Please mind, however, that a precise critical analysis of the concepts and methods is very difficult due to the variety of adsorbents used, the methods of their application in various pathological conditions, and frequently, an insufficient amount of information (e.g. in the case of per rectum application). Thus, we had to retain some of the generalizations as they are presented in the descriptions. We hope this is acceptable.
- Conclusions need to be critical and specific. It needs to highlight the achievements and specific scientific gaps in our knowledge. So what further research should have priority?
Answer: This is a good point – thank you! We re-shaped the Conclusions so they feature those elements. Table 3 is made and pointed out here.
- Lot of corrections are required in the reference. Carefully revise it.
Answer: We apologize for those errors! We made the best effort to rectify them all.
Authors

Reviewer 2 Report
This review provides a summary of the current knowledge regarding activated carbon in various inflammatory conditions, including sepsis. It highlights the mechanism of action of activated carbon, its ability to adsorb harmful substances, and its potential benefits in reducing inflammation and supporting homeostasis. This paper also mentions the advantages of activated carbon in terms of microbiota restoration, reduction of oxidative stress, and its role in accelerating bone marrow restoration. Additionally, it discusses the importance of early administration of activated carbon to prevent irreparable damage caused by dysregulated PAMPs and DAMPs. This article also suggests potential applications of activated carbon in other inflammatory states and explores the use of carbon nanoparticles as a promising alternative. Overall, the review seems to provide a comprehensive overview of the current understanding of activated carbon as a therapeutic agent in inflammatory conditions. In my opinion it also provides a balanced view of the topic thus it can be considered as a valuable work. The literature items used in this work have been meticulously selected to align with the topic, and they are all up-to-date, ranging from the past 2 to 5 years. Reading this manuscript has been an absolute delight due to its unique perspective and the meticulous preparation that has gone into it.
Author Response
09/07/2023
Dear Reviewer,
We wish to resubmit the review entitled “Activated carbon for sepsis prevention and intervention: a modern way of utilizing old therapies.” following revision for consideration by the Journal of Carbon Research in the special issue Carbon for Health and Environmental Protection.
Thank you for your work and appreciation of our manuscript.
Authors

Reviewer 3 Report
Activated carbon for sepsis prevention and intervention: a modern way of utilizing old therapies.
This work is a significant contribution to the field because uncontrolled inflammation leads contributes to life-threatening sepsis complications such as multi-organ dysfunction syndrome (MODS) including abnormal levels of pathological and damage-associated molecular patterns (PAMPs & DAMPs) in their biological fluids.
Clinical and laboratory investigations demonstrated that activated carbon or charcoal (AC) mitigates acute inflammation when applied orally, per rectum or in a hemoperfusion system. The mechanism of AC acting through an unspecific (broad-range) or/and specific adsorption of PAMPs& DAMPs from biological fluids. AC has an anti-inflammatory, antioxidant, as well as microbiome- bone marrow-protecting agent with high adsorptive being considered an adjunctive treatment for preventing sepsis and/or multi-organ failure. Article which includes a figure and two tables is well organized and comprehensively described. There are appropriate and adequate references to related and previous work.
Author Response
Dear Reviewer,
We wish to resubmit the review entitled “Activated carbon for sepsis prevention and intervention: a modern way of utilizing old therapies.” following revision for consideration by the Journal of Carbon Research in the special issue Carbon for Health and Environmental Protection.
Thank you for your work and favorable reviews of our manuscript.
Authors
